# Compact GSPN: Scaling Spatial Propagation to Vision Foundation Models

## Abstract

Scaling vision foundation models is limited by the quadratic cost of self-attention. Generalized Spatial Propagation Networks (GSPN) provide a linear-time alternative that propagates context directly on the 2D grid and removes positional embeddings, but have not been scaled to foundation-level training. We present Compact GSPN (C-GSPN), a ViT block with a compressed propagation space that preserves accuracy while cutting propagation latency by nearly $10\times$, complemented by lightweight projections and fused CUDA kernels for further efficiency. To pretrain at scale, we use a two-stage distillation scheme with module-wise supervision and end-to-end alignment. In a representative 1K configuration (batch 32, $C{=}1152$), C-GSPN yields up to $2\times$ speedup, while maintaining competitive zero-shot accuracy and improving segmentation by $+2.1\%$. Extensive experiments and ablations confirm that the proposed compression and two-stage distillation are key to achieving strong transfer while substantially reducing compute, offering a practical path toward subquadratic vision foundation models.

## 1 Introduction

Scaling up vision foundation models is increasingly constrained by the cost of self-attention (Vaswani et al., 2017). As input resolution grows, token counts explode, and attention's quadratic complexity quickly becomes the dominant cost for both memory and latency. This scalability bottleneck limits the practical use of high-resolution inputs and slows down training and inference for large models.

A rich line of work has tried to address this by making attention subquadratic—through token sparsity (Child et al., 2019), local windows (Dai et al., 2019), or kernelized approximations (Choromanski et al., 2020)—reducing computation while attempting to preserve global context. These approaches boost throughput but still struggle to provide a consistently good accuracy–latency trade-off, often requiring careful tuning or additional components.

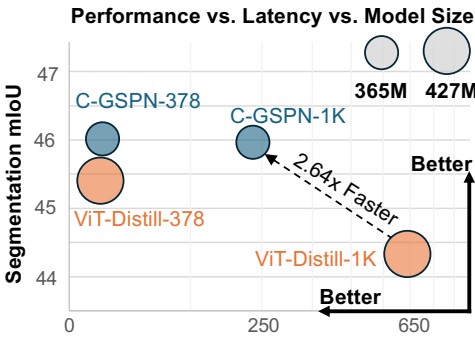

Figure 1: Comparison of C-GSPN (ours) vs. ViT-Distill on dense prediction at 378 and 1K resolutions; lower latency and higher accuracy are better.

Generalized Spatial Propagation Networks (GSPN) (Wang et al., 2025) offer an alternative: they propagate information directly over the 2D grid, capture long-range context with linear complexity, and eliminate the need for positional embeddings. Despite these advantages, GSPN has not yet been scaled to the data and model sizes of modern vision foundation models, leaving open the question of how to pretrain such architectures effectively at foundation scale.

In this work, we study how to scale GSPN to a vision foundation model. We introduce Compact GSPN (C-GSPN), a block that replaces the attention sublayer with a compressed Generalized Spatial Propagation Network. Unlike attention, which aggregates affinities across channels, GSPN propagates features independently per channel. Theoretically, propagation time remains flat if the number of channels stays below the GPU's concurrency threshold; beyond this, runtime grows roughly linearly with the channel dimension due to hardware limits (see Fig. 2). To control this growth, we

project the propagation into a compressed latent space that preserves accuracy while cutting the cost of the propagation step, yielding nearly $10\times$ latency reduction at the layer-level, without accuracy degradation. Building on this, we further reduce overhead from surrounding operators through two key refinements: a lightweight redesign of the Q/K/V projection and a fused CUDA kernel that lowers the cost of non-linearities such as sigmoid. Together, these improvements form a compact yet faithful substitute for attention within a ViT block. Compared to the original GSPN, this compact design yields large practical speedups (e.g., $13.7\times$ at 1K resolution). Against softmax attention—even with highly optimized FlashAttention (Dao et al., 2022)— C-GSPN reduces block latency by $2\times$ at 1K and $4\times$ at 2K resolution, positioning it as a competitive alternative for foundation-scale encoders.

To retain model quality, we introduce a two-stage cross-operator distillation recipe. In the first stage, we perform sublayer-wise supervision that aligns intermediate representations of each GSPN block with its attention-based teacher block. In the second stage, we fine-tune the entire network end-to-end. On top of this, we adopt a sparse feature distillation strategy that supervises two anchor points—Post-Propagation and Post-Block—providing both local and block-level guidance. Distilling from a SigLIP-v2 (Tschannen et al., 2025) teacher on 600M image–text pairs, the resulting C-GSPN encoder achieves $81.3\%$ (vs. $82.2\%$ baseline) and improves segmentation accuracy by $+2.1\%$ on ADE20K. Finally, we leverage C-GSPN's inherent spatial structure for efficient resolution transfer, enabling smooth adaptation across resolutions—from $378\times378$ to $1036\times1036$—using just $1/200$ of the data required for training from scratch. This property makes its efficiency advantage even more pronounced at higher resolutions.

We evaluate C-GSPN against established vision foundation models across multiple downstream tasks. In particular, we conduct ablations with a SigLIP-v2 teacher to analyze: (i) the impact of latent compression on accuracy–latency trade-offs, (ii) the effectiveness of two-tap supervision (post-propagation vs. post-block), and (iii) high-resolution transfer without positional embeddings. These studies show that C-GSPN achieves superior computational efficiency while maintaining competitive performance across diverse vision tasks.

**Contributions.** (1) We propose C-GSPN, a compressed spatial propagation block that serves as a drop-in, positional-embedding-free substitute for attention in ViTs, scaling linearly with resolution. (2) We design a compute-aware implementation combining latent-space propagation and a fused CUDA normalization kernel that substantially reduces latency at high resolution. (3) We introduce a progressive, cross-operator distillation recipe with two supervision taps per block and lightweight feature adaptors, enabling effective transfer from attention to propagation. (4) We demonstrate practical high-resolution encoder transfer without tiling or positional embeddings via curriculum learning with upsampling self-distillation, yielding strong dense-task performance.

## 2 RELATED WORK

Due to space limits, we focus on subquadratic alternatives to full softmax attention; a broader survey appears in the Appendix.

**Subquadratic Attention and Alternatives.** Subquadratic alternatives to full softmax attention aim to reduce the $\mathcal{O}(N^2)$ dependence on token count while preserving global interactions. *Sparsity- and window-based* designs such as Longformer and BigBird in NLP and Swin for vision constrain attention to local windows with a few global tokens, yielding near-linear scaling but making long-range mixing sensitive to the chosen sparsity pattern and hyperparameters (Beltagy et al., 2020; Zaheer et al., 2020; Liu et al., 2021). *Kernelized/low-rank* approaches linearize attention—e.g., Linear Transformers, Performer, Nyströmformer, Linformer—trading exactness for approximation; their accuracy often depends on the feature map, rank, or landmark scheme and requires careful tuning (Katharopoulos et al., 2020; Choromanski et al., 2020; Xiong et al., 2021; Wang et al., 2020). IO-aware exact attention like FlashAttention reduces constant factors via optimized memory access, yet latency still scales quadratically with tokens at high resolution (Dao et al., 2022). Beyond attention, *state-space models* (S4; Mamba) offer linear-time sequence operators, but adapting 1D formulations to high-resolution vision typically requires extra 2D inductive bias or hierarchical designs (Gu et al., 2021; Gu & Dao, 2023). In contrast, *spatial propagation networks* operate natively on 2D grids and remove positional embeddings; recent GSPN extends this idea to four-direction propagation with linear complexity (Liu et al., 2017; Wang et al., 2025). Our work scales GSPN to foundation-model pretraining through a compact, CUDA-optimized instantiation distilled from ViT teachers, achieving substantially lower latency at 1K–2K while maintaining competitive transfer.

## 3 BACKGROUND

To facilitate a clear understanding of our C-GSPN approach, we start with an overview of the 2D Spatial Propagation architecture as the foundation of our C-GSPN model. Then, we provide the GPU Hardware Capabilities and the Kernel Execution of 2D Linear Propagation to understand the performance of our C-GSPN model.

**2D Linear Propagation Algorithm.** Spatial propagation (Wang et al., 2025; Liu et al., 2017) provides a linear alternative to attention by propagating features along four directions of the 2D grid. For an input $x \in \mathbb{R}^{H \times W \times C}$, the hidden state $h$ with the same dimension is computed sequentially across one dimension (e.g., row by row), while all positions within each row are updated in parallel. Taking the top-to-bottom pass as an example, with $i \in [0, H-1]$ and channel $c$, let $h_{i,:,c}, x_{i,:,c} \in \mathbb{R}^W$, $\lambda_{i,:,c} \in \mathbb{R}^W$, and $w_{i,c} \in \mathbb{R}^{W \times W}$. The recurrence is:

$$h_{i,:,c} = w_{i,c} \, h_{i-1,:,c} + \text{Diag}(\lambda_{i,:,c}) \, x_{i,:,c} \tag{1}$$

with $h_{0,:,c}$ initialized from $x_{0,:,c}$.

The final output for row $i$ and channel $c$ is given by $y_{i,:,c} = u_{i,:,c} \odot h_{i,:,c}$. All parameters $\lambda$, $w$, and $u$ are input-dependent.

To satisfy the Stability–Context Condition (Wang et al., 2025), each $w_{i,c}$ is row-stochastic (rows sum to 1). We parameterize this by normalizing the nonzero connections within the neighbor set $\mathcal{N}(j)$ for position $j$ in row $i$:

$$w_{i,c}(j, k) = \frac{\sigma(\tilde{w}_{i,c}(j, k))}{\sum_{k' \in \mathcal{N}(j)} \sigma(\tilde{w}_{i,c}(j, k'))} \tag{2}$$

Propagation along four directions—top-to-bottom, bottom-to-top, left-to-right, and right-to-left—produces dense pairwise connectivity with only three coefficients per pixel per pass. In the tridiagonal case ($\mathcal{N}(j) = \{j-1, j, j+1\}$), this reduces to three nonzero entries per row, normalized locally.

A single top-to-bottom pass performs $O(H)$ sequential steps while all $W$ elements of each row are computed in parallel (symmetrically $O(W)$ for a column pass). Running both row- and column-wise scans gives an effective sequential depth of $O(\max(H, W))$, i.e., $O(\sqrt{N})$ for a square map with $N = HW$ pixels. We describe the preliminary of GPU Hardware and Kernel Execution of GSPN in the Appendix.

## 4 METHODOLOGY

We align C-GSPN closely with the ViT structure and adopt consistent terminology: (i) *Sublayer*: the latent-space 2D propagation unit, analogous to the scaled dot-product attention sublayer, (ii) *Layer*: a full C-GSPN layer (Fig. 4), which parallels a multi-head attention layer, and (iii) *Block*: a Transformer block where the attention layer is replaced by a C-GSPN layer (Fig. 8). In Sec. 4.1, we discuss how to improve the *sublayer* efficiency of a ViT block; then in Sec. 4.1, we show how to reduce the overhead of a full C-GSPN layer with respect to the non-propagation components.

### 4.1 EFFICIENCY BOOST WITH LATENT SPACE AND CUDA KERNEL

Original GSPN sublayers propagate features independently on each channel $C$ of $\mathbf{x} \in \mathbb{R}^{B \times C \times H \times W}$, performing four directional line scans as in Eq. 1. Since modern GPUs support only limited resident blocks per SM with constrained registers, when either the batch size $B$ or channel dimension $C$ grows, excess slices serialize, causing latency spikes despite theoretical parallelism (see Appendix). Figure 2 illustrates this effect: latency remains flat at small $B/C$, but jumps sharply once concurrency saturates (e.g., $11.57\times$ increase when $C$ grows from 288 to 576, and $7.76\times$ when $B$ grows from 8 to 16).

**Latent-space 2D propagation.** To avoid hitting the concurrency wall, we move the propagation into a compressed latent space. Let $s > 1$ be a compression factor and compressed channel number

$C_c = \lfloor C/s \rfloor$. We introduce linear projections $P_\downarrow : \mathbb{R}^C \to \mathbb{R}^{C_c}$ and $P_\uparrow : \mathbb{R}^{C_c} \to \mathbb{R}^C$ applied per spatial location (i.e., $1 \times 1$ convolutions across channels):

$$\mathbf{x}_c = P_\downarrow(\mathbf{x}) \in \mathbb{R}^{B \times C_c \times H \times W} \tag{3}$$

We generate propagation parameters directly in the latent channel space:

$$u = L_u(\mathbf{x}_c), \quad \lambda = L_\lambda(\mathbf{x}_c), \quad \tilde{w} = L_w(\mathbf{x}_c) \tag{4}$$

where $u, \lambda \in \mathbb{R}^{B \times C_c \times H \times W}$ and $\tilde{w} \in \mathbb{R}^{B \times C_c \times H \times W \times 3}$. $L_u$, $L_\lambda$, and $L_w$ denote learned linear heads implemented as $1 \times 1$ convolutions over channels (applied at each spatial position) that predict per-position parameters.

For a top-to-bottom propagation and any compressed channel $\tilde{c} \in \{1, \dots, C_c\}$, the per-row recurrence mirrors Eq. 1 but operates entirely in the latent channels using Eq. 4 and Eq. 2 (tridiagonal specialization):

$$h_{i,:,\tilde{c}} = w_{i,\tilde{c}} \, h_{i-1,:,\tilde{c}} + \mathrm{Diag}(\lambda_{i,:,\tilde{c}}) \, x_{c,\,i,:,\tilde{c}}, \quad y_{i,:,\tilde{c}} = u_{i,:,\tilde{c}} \odot h_{i,:,\tilde{c}} \tag{5}$$

Here $w$ is normalized row-stochastically as in Sec. 3. We run the four directional scans in the latent space and then up-project only once at the end:

$$\mathbf{y}_c = \mathrm{Prop}_{2D}(\mathbf{x}_c; u, \lambda, w), \quad \mathbf{y} = P_\uparrow(\mathbf{y}_c) \in \mathbb{R}^{B \times C \times H \times W} \tag{6}$$

This reformulation reduces the effective grid size from $B \times C$ to $B \times C_c$ (with $C_c = \lfloor C/s \rfloor$), lowering per-SM pressure and avoiding serialization. Empirically at 1K resolution, latency remains nearly flat across channels and batch sizes; compared to the raw-space kernel, latent-space propagation attains $54.46\times$ speedup at $C{=}1152$ and $55.74\times$ at $B{=}32$ (Fig. 2).

Importantly, because $\tilde{w}$ are defined in the latent channels, the row-stochastic normalization of Eq. 2 is evaluated over $C_c$ rather than $C$, yielding an additional $38.9\times$ speedup for weight normalization.

**Non-GSPN Overhead Reduction.** Is the propagation sublayer really the bottleneck? For softmax attention at high resolution, yes. But for GSPN—already efficient—the overhead of non-propagation parts dominates, especially at low to mid resolutions. At 1K, these non-propagation parts cost $9.6\times$ more than the core propagation (Fig. 3). To address this, we remove: (i) the inner-module residual path around the propagation kernel, (ii) the linear projections inherited from the attention-based template, and (iii) the intermediate upsample projections that previously expanded channels before propagation. Cumulatively, these edits yield a $\sim 5.5\times$ reduction in overhead latency (Fig. 3).

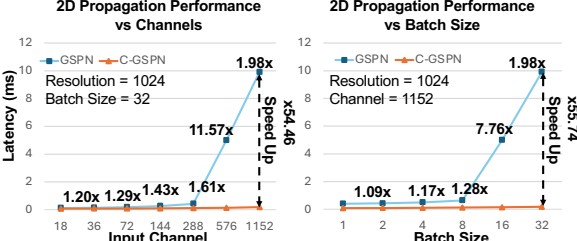

Figure 2: Original GSPN vs C-GSPN Propagation sublayer latency under increasing channels (left) and batch size (right) at 1K resolution. Original GSPN exhibits spikes as $C/B$ grow due to GPU concurrency limits; C-GSPN remains flat, yielding large speedups.

**Fused CUDA Normalization.** We optimize the row-stochastic normalization step (Sec. 3) by fusing its sequence of operations—sigmoid activation, local reduction, clamping, and division—into a single custom CUDA kernel. By executing all steps in one pass, this eliminates intermediate memory traffic and kernel launch overhead, achieving a $2.15\times$ speedup over PyTorch's baseline. Combined with latent-space structural reduction ($C \to C_c$; e.g., $1152 \to 64$), the effective cost of normalization is reduced by $83.68\times$ at 1K resolution.

Overall, these three improvements yield a $13.7\times$ speedup of the GSPN layer at 1K resolution. Thorough comparisons are reported in Sec. 5.

### 4.2 SCALING C-GSPN VIA DISTILLATION

Although GSPN Wang et al. (2025) has shown strong performance on mid-scale tasks such as image classification and generation, its usage in foundation-scale vision models remains underexplored.

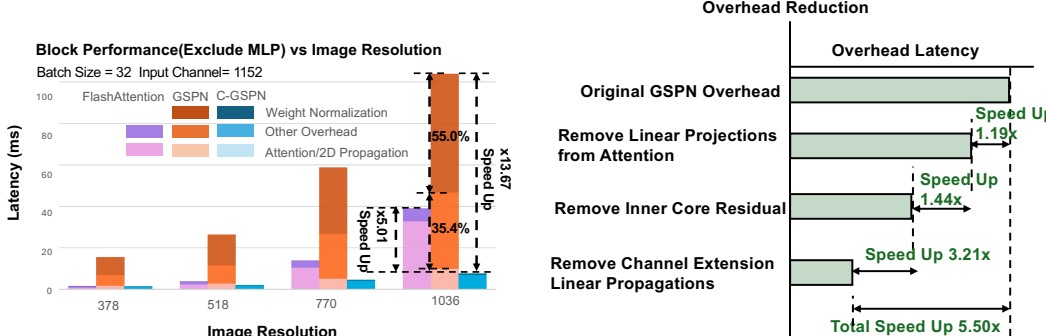

Figure 3: **Left**: Block latency vs. image resolution (B=32, C=1152), where the original GSPN is dominated by weight normalization and other overhead at high resolution; C-GSPN substantially reduces both. **Right**: Overhead reduction at 1K resolution from removing (1) Additional linear projections inherited from attention, (2) inner-module residual, and (3) Channel Extension projections; cumulative speedup about $\times 5.5$.

We scale C-GSPN to this regime by aligning its block design with a SigLIP 2-style ViT Zhai et al. (2023), a widely used architecture for efficient contrastive pretraining and robust benchmarking. Training such models from scratch is computationally prohibitive, so we adopt knowledge distillation: a pre-trained quadratic-attention teacher supervises a GSPN-based student. However, cross-operator transfer is non-trivial: attention mixes tokens via explicit pairwise interactions, whereas GSPN attains global context through sequential local propagation that reduces the effective sequence length to $\sqrt{N}$ for $N$ elements. This mismatch makes direct attention layer weight transfer inappropriate and induces a feature-distribution gap that must be handled explicitly. In the following, we address this gap with a progressive, two-stage distillation strategy that first aligns intermediate features block by block, and then fine-tunes the full model end-to-end.

**Stage 1: Sublayer-wise Pretraining.** As shown in Fig. 8, we begin by aligning each C-GSPN propagation sublayer with its corresponding attention sublayer in the teacher. For each block $i$, both teacher and student take the output of the $(i-1)$-th teacher block as input:

$$\mathbf{h}^{t,(0)} = \mathbf{x}, \qquad \mathbf{h}^{t,(i)} = \text{TeacherBlock}^{(i)}\big(\mathbf{h}^{t,(i-1)}\big). \tag{7}$$

Given this shared input $\mathbf{h}^{t,(i-1)}$, we compute the sublayer features:

$$F^{s,(i)} = f^{(i)}_{\text{C-GSPN-prop}}\big(\mathbf{h}^{t,(i-1)}\big), \qquad F^{t,(i)} = f^{(i)}_{\text{Attention}}\big(\mathbf{h}^{t,(i-1)}\big), \tag{8}$$

where $F^{s,(i)}$ and $F^{t,(i)}$ denote outputs immediately after the student's propagation sublayer and the teacher's attention sublayer, respectively. We minimize a simple feature alignment loss:

$$\mathcal{L}^{(i)}_{\text{prop}} = |F^{s,(i)} - F^{t,(i)}|^2_2. \tag{9}$$

The teacher is frozen, and gradients flow only through the student sublayer. Importantly, each block is trained independently without backpropagation across blocks, so every C-GSPN sublayer directly learns to mimic the representational pattern of its paired attention sublayer. This parallel scheme stabilizes training and provides a strong initialization for subsequent end-to-end distillation.

**Stage 2: End-to-end Distillation.** After layer-wise pretraining, we optimize end-to-end with *two supervision taps per block*. For clarity, we refer to the feature taken *after the propagation/attention sublayer* as post-propagation (PP) and the feature taken *after the entire block (prop/attn + MLP + norms)* as post-block (PB). The rationale is to *decompose* cross-operator transfer: PB supervision preserves the teacher's block transformation, where the MLP is largely isomorphic across student/teacher, while PP supervision directly pressures the GSPN sublayer to learn the teacher's attention-style mixing, rather than letting the MLP "absorb" the mismatch.

Let $V^{s/t}_{\text{PP}}$ and $V^{s/t}_{\text{PB}}$ denote student/teacher features at PP and PB, and let $P(\cdot)$ be token-wise softmax. We use MSE loss for feature alignment and Kullback–Leibler divergence (KL) loss for distribution matching:

$$\mathcal{L}_{\text{PP}} = \text{MSE}\big(V^s_{\text{PP}}, V^t_{\text{PP}}\big) + \lambda_1 \text{KL}\big(P(V^s_{\text{PP}}) \,\|\, P(V^t_{\text{PP}})\big), \tag{10}$$

$$\mathcal{L}_{\text{PB}} = \text{MSE}\big(V^s_{\text{PB}}, V^t_{\text{PB}}\big) + \lambda_2 \text{KL}\big(P(V^s_{\text{PB}}) \,\|\, P(V^t_{\text{PB}})\big).$$

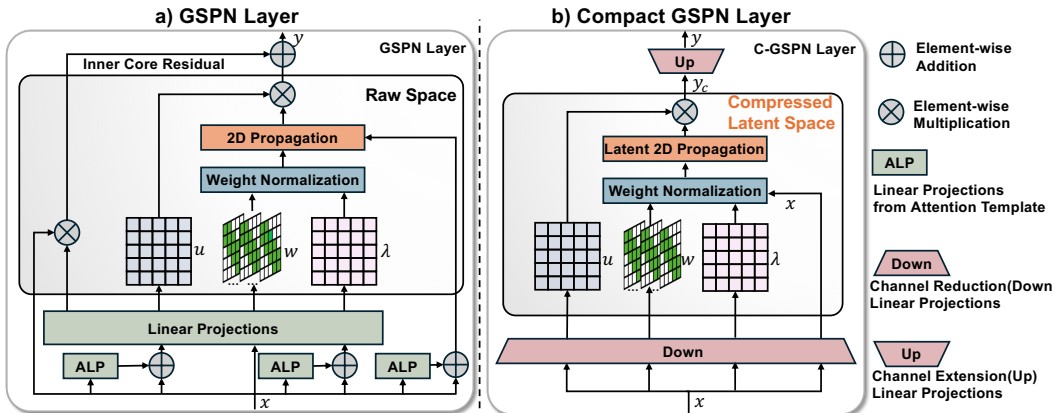

Figure 4: **Layer comparison of GSPN and C-GSPN.** Left: Original GSPN operates in raw channel space and retains extra projections and residuals inherited from attention. Right: C-GSPN introduces latent-space propagation and fused normalization, while removing redundant projections and residuals, resulting in a lighter and faster design.

We insert lightweight adaptors before each tap when needed to reduce feature-space mismatch, which is discussed in the following. This dual-tap design regularizes cross-operator distillation by giving the GSPN sublayer a dedicated teacher target (PP) while keeping block-level alignment (PB), which we validate via ablations in Sec. 5.3. Similar principles—tap supervision for ViTs and staged cross-architecture alignment—have been validated in recent knowledge distillation work (Touvron et al., 2021; Yang et al., 2022; Bick et al., 2024).

**Feature Adaptor for Distillation.** Even with dual supervision, directly matching C-GSPN and ViT features is difficult because the two architectures compute representations in fundamentally different ways: spatial propagation aggregates context sequentially, whereas attention mixes all tokens at once. This mismatch leads to unstable training if features are compared in raw space.

To address this, we introduce lightweight *feature adaptors* that act as learnable bridges between student and teacher features. As shown in Fig. 8, adaptors are inserted before the distillation losses at both taps. At the post-propagation (PP) tap, the adaptor maps the raw GSPN output $V_{PP}^s$ into an aligned representation $\hat{V}_{PP}^s$; at the post-block (PB) tap, it transforms the student block output $V_{PB}^s$ into $\hat{V}_{PB}^s$. The distillation objectives then become:

$$\mathcal{L}_{\text{PP}} = \text{MSE}(\hat{V}_{PP}^s, V_{PP}^t) + \lambda_1 \text{KL}(P(\hat{V}_{PP}^s)\|P(V_{PP}^t)), \tag{11}$$

$$\mathcal{L}_{\text{PB}} = \text{MSE}(\hat{V}_{PB}^s, V_{PB}^t) + \lambda_2 \text{KL}(P(\hat{V}_{PB}^s)\|P(V_{PB}^t)),$$

and the total objective is

$$\mathcal{L}_{\text{total}} = \alpha \, \mathcal{L}_{\text{PP}} + \beta \, \mathcal{L}_{\text{PB}}. \tag{12}$$

By transforming the supervision task from direct feature matching to learnable feature alignment, adaptors ease cross-operator transfer. In practice, they stabilize optimization at PP (where the operator gap is largest) and yield consistent improvements in downstream accuracy (See ablations in Sec. 5.3).

Furthermore, inspired by the hybrid Mamba-Transformer design in MaTVLM (Li et al., 2025), which demonstrates that a balanced integration of sequential state-space models with attention mechanisms yields superior performance over pure architectures, we similarly observe that allocating a small attention budget results in a better accuracy-latency trade-off compared to either pure attention or pure C-GSPN. With this insight, we adopt a hybrid architecture that preserves a modest fraction of attention layers while incorporating C-GSPN blocks. Further details on our implementation and empirical validation are provided in Sec. 5 and Appendix.

## 4.3 HIGH-RESOLUTION ENCODER DISTILLATION

High-resolution in downstream tasks is often handled with tiling because attention cost grows quadratically with resolution; however, tiling increases engineering complexity, introduces bound-

ary artifacts, and sacrifices global context. By contrast, C-GSPN maintains low latency at 1K–2K resolution and supports single-pass inference without tiling (Sec. 5.1), which enables high-resolution encoders. Importantly, C-GSPN requires no positional embeddings, so moving to higher resolutions does not involve modifying the architecture—only adapting training.

We therefore study how to transfer low-resolution checkpoints to higher resolutions under limited compute. Two challenges emerge. First, naively transferring from a base resolution $r_0$ (e.g., 378) to a target resolution $r_K$ (e.g., 756) yields suboptimal performance. A curriculum learning strategy (Bai et al., 2023; Chen et al., 2023b;a; Li et al., 2024) that gradually increases resolution ($378 \rightarrow 518 \rightarrow 756$) significantly improves results ($80.4\%$ vs. $70.2\%$ with equal sample budgets), showing that progressive scaling stabilizes adaptation. Second, contrastive objectives alone provide sufficient supervision for classification but fail to capture the fine-grained spatial detail needed in dense tasks like segmentation (Sec. 5).

To address these challenges, we propose to combine curriculum learning with upsampling self-distillation. At each step $k > 0$, the model at resolution $r_{k-1}$ serves as a frozen teacher, whereas its features are bilinearly upsampled to $r_k$ and guide the student at both module and block levels using the dual objectives from Sec. 4.2:

$$\tilde{V}_m^{t,(k)} = \text{Up}\left(V_m^{t,(k-1)}\right), \qquad \tilde{V}_b^{t,(k)} = \text{Up}\left(V_b^{t,(k-1)}\right), \tag{13}$$

$$\mathcal{L}_{\text{hr}}^{(k)} = \alpha\,\mathcal{L}_{\text{module}}^{(k)} + \beta\,\mathcal{L}_{\text{block}}^{(k)}, \tag{14}$$

where $\text{Up}(\cdot)$ denotes bilinear upsampling from resolution $r_{k-1}$ to $r_k$; $\mathcal{L}_{\text{module}}^{(k)}$ and $\mathcal{L}_{\text{block}}^{(k)}$ follow the same MSE + KL formulation as in Sec. 4.2, applied to the upsampled teacher features $\tilde{V}_m^{t,(k)}$, $\tilde{V}_b^{t,(k)}$ and student features $V_m^{s,(k)}$, $V_b^{s,(k)}$ at resolution $r_k$. Despite using approximate supervision, Table 2 shows that it substantially improves dense-task performance, enabling the student to align feature distributions while preserving C-GSPN's global context modeling.

## 5 Experiments

We evaluate C-GSPN along two axes: *system efficiency*-latency of the core sublayer and the full Transformer block across resolutions, and *model quality* at foundation scale (zero-shot transfer and dense tasks), followed by ablations and high-resolution transfer under limited compute.

### 5.1 System Efficiency (Latency/Throughput)

We benchmark C-GSPN against attention mechanisms and original GSPN across varying resolutions, evaluating both the propagation sublayer and complete ViT blocks.

**Experimental Setup.** We report latency on A100 GPUs with batch size 32 and 1152 channels, sweeping input side length from 378 to 2058. We compare four cores—standard attention, FlashAttention, original GSPN, and C-GSPN—and their corresponding *blocks* (core + MLP, norms, residuals). Tiled FlashAttention is included as a high-resolution baseline.

**Sublayer Results.** Fig. 5 (top) shows dramatic scaling differences. Original GSPN remains $67.2\times$–$86.9\times$ slower than C-GSPN at 1K and 2K. FlashAttention requires 500 ms per layer at 2K, but C-GSPN maintains just 0.462 ms, providing a $1000\times$ **speedup** over FlashAttention.

**Block Results.** The complete block comparison (Fig. 5, bottom) includes MLP, normalization, and residual connections. While FlashAttention blocks require over 600ms at 2K resolution, C-GSPN blocks complete in under 150ms, yielding a **$4\times$ end-to-end speedup** at 2K.

**Tiling Replacement.** To manage computational constraints, attention models often rely on tiling to fit high-resolution inputs, but this adds complexity, coordination overhead, and boundary artifacts. In contrast, C-GSPN processes inputs in a single pass without tiling, outperforming tiled FlashAttention: **$78.48\times$ faster** at the core sublayer and $10\%$ faster at the block level for 2K resolution. This enables seamless ultra-high-resolution processing while preserving global context, which is especially valuable for dense prediction and high-resolution analysis tasks.

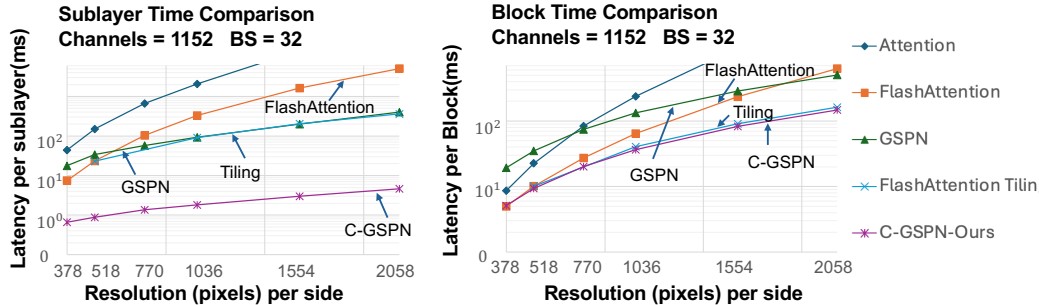

Figure 5: **Latency comparison across resolutions.** Left: Sublayer latency (attention vs. C-GSPN) in log scales. Right: Full block latency. C-GSPN maintains low, resolution-stable latency while attention methods scale quadratically. At 2K, C-GSPN outperforms FlashAttention $1000\times$ at the sublayer and $4\times$ at the block level, while still surpassing tiled FlashAttention $78.48\times$ at the sublayer and $10\%$ fasterat the block level.

## 5.2 PERFORMANCE EVALUATION OVER VISION TASKS

We next evaluate C-GSPN's end-to-end model quality at foundation scale, complementing the system-level efficiency in Sec. 5.1. We compare against two student baselines trained under the same data/budget: (i) an *isomorphic* ViT→ViT student (identical block topology and parameter shapes), and (ii) an *original* GSPN student. All students are distilled from a strong contrastive ViT teacher (OpenCLIP ViT-SO/14 at 378) (Radford et al., 2021; Ilharco et al., 2021), and evaluated on zero-shot classification (ImageNet Top-1/Top-5) (Russakovsky et al., 2015), dense segmentation (ADE20K-F, ADE20K, PASCAL) (Zhou et al., 2019; Everingham et al., 2010), and object detection (COCO) (Lin et al., 2014). For context, we also report large-scale pretrained variant (OpenCLIP), which served as the teacher model.

**Results.** As summarized in Table 1, C-GSPN uses $15\%$ fewer parameters yet nearly matches the ViT→ViT student on the macro average (63.3 vs. 63.5), while outperforming the original GSPN student across all reported metrics and achieving stronger segmentation scores than the teacher. Coupled with Sec. 5.1, C-GSPN preserves its large efficiency advantage at high resolution, delivering up to **3.3× end-to-end network speedup** at 2K and enabling single-pass (no-tiling) inference—particularly beneficial for dense prediction.

## 5.3 ABLATION STUDIES

**Training Strategy.** We ablate the supervision scheme from Sec. 4.2 cumulatively: (i) contrastive-only baseline; (ii) + PB loss (post-block; features after the entire block); (iii) + a lightweight 2-layer MLP adaptor at the taps to reduce feature-space mismatch; (iv) + PP loss (post-propagation; features after the propagation sublayer); (v) + Stage-1 sublayer-wise pretraining. Fig. 6a shows monotonic gains at each step: the largest jump comes from PP supervision (direct signal to the propagation sublayer); the adaptor provides steady improvements by aligning feature spaces; and sublayer-wise pretraining yields a strong initialization that persists through end-to-end training.

**Module Structure.** We compare structural variants (Fig. 6b): C-GSPN with compression ratios 12/18/72, pure 2D propagation, and hybrid variants that replace a small subset of propagation layers with attention (3 out of 27 in experiments). We discovered two interesting findings: (i) Lower compression (more latent channels) improves representational capacity up to a point; reducing compression further does not yield additional accuracy gains. Under a fixed budget, C-GSPN-18 provides the best accuracy–efficiency balance. (ii) Replacing 3 out of 27 layers ($1/9$) with attention yields consistent gains over pure C-GSPN. The intuition is targeted: attention is used sparingly to inject long-range pairwise mixing in a few layers, while the remaining layers retain efficient global propagation. This avoids quadratic cost throughout the network yet improves accuracy, and is gaining popularity in recent works (Waleffe et al., 2024; Dong et al., 2025; Basant et al., 2025). At a compression ratio of 18, the hybrid variant both improves accuracy over its pure counterpart and remains $2.4\times$ faster at 1K and $3.3\times$ faster at 2K compared to pure-attention baselines.

## 5.4 HIGH-RESOLUTION TRANSFER EXPLORATION

C-GSPN's single-pass efficiency enables compute-aware transfer to higher resolutions without tiling. Instead of costly full-scale training (600M samples), we adopt a lightweight resolution curriculum of 3M samples (1M per stage), scaling $378 \rightarrow 518 \rightarrow 756 \rightarrow 1036$. At each stage, the

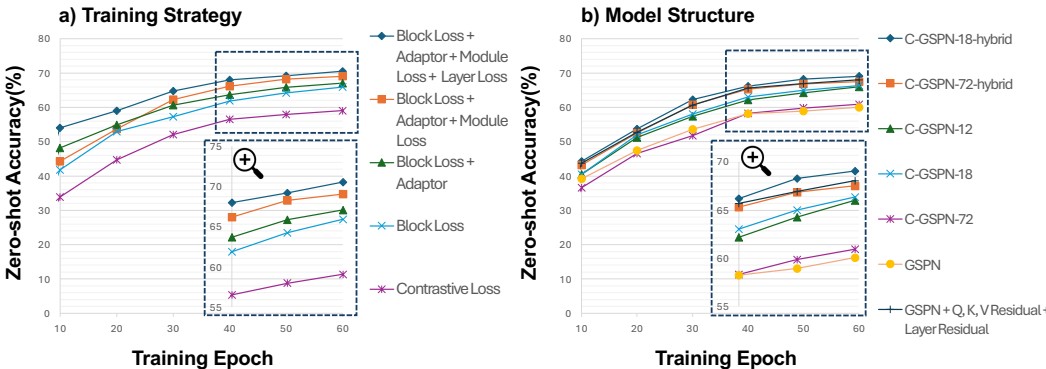

Figure 6: **Ablations on training strategy and module structure.** (a) Cumulative strategy: the module-level distillation provides the largest jump by directly supervising 2D propagation; the adaptor mitigates feature mismatch; layer-wise pretraining gives a strong start. (b) Structure: moderate compression (ratio 18) strikes the best accuracy–efficiency balance; a small attention budget (3 of 27 layers) further improves accuracy while preserving speed.

| Method | Params. | Res. | Patches | Classification | | Segmentation | | | Detection | Avg. |
|---|---|---|---|---|---|---|---|---|---|---|
| | | | | Top-1 | Top-5 | ADE20K-F | ADE20K | Pascal | COCO | |
| OpenCLIP SO/14 | 427M | 378 | 729 | 84.1 | 97.4 | 42.8 | 45.8 | 77.5 | 47.7 | 64.6 |
| ViT-Distill | 427M | 378 | 729 | **82.2** | **96.7** | 43.2 | 45.5 | 77.2 | **45.8** | 63.5 |
| GSPN | 477M | 378 | 729 | 80.5 | 95.8 | 44.3 | 45.3 | 77.2 | 44.3 | 62.7 |
| **C-GSPN (ours)** | 365M | 378 | 729 | 81.3 | 96.3 | **44.7** | **46.0** | **77.6** | 45.0 | 63.3 |

Table 1: **Comprehensive evaluation across vision tasks.** OpenCLIP SO/14 is the teacher for distilled models. We report classification, segmentation, and detection metrics alongside parameters (Params.), resolution (Res.), and number of patches. Average (Avg.) is a macro average over tasks: mean(mean(Top-1, Top-5), mean(ADE20K-F, ADE20K, Pascal), COCO). AED20K-F uses feature tokens as in EfficientViT (Cai et al., 2023); ADE20K uses both feature and summary tokens as in TIPS (Maninis et al., 2025).

| Resolution | 378 | 518 | 756 | 1036 | Latency(1K) |
|---|---|---|---|---|---|
| ViT-Distill | 45.5 | – | – | 44.1 | 633.6(s) |
| C-GSPN w/o KD | 46.0 | 45.1 | 44.5 | 43.5 | 242.4(s) |
| C-GSPN w/ KD | 46.0 | 46.3 | 46.2 | 45.8 | (2.64× Speed up) |

Table 2: **High-resolution transfer under limited compute.** We report segmentation accuracy (ADE20K) across increasing input resolutions. KD indicates knowledge distillation. We also report single-GPU inference latency at 1036 resolution per 1000 samples with batch size 1. C-GSPN yields a 2.64× speedup.

previous checkpoint is frozen as a teacher whose post-propagation and post-block features supervise the next resolution via MSE+KL (See Sec. 4.3). As shown in Table 2, this staged self-distillation yields gains on dense tasks segmentation at 518 resolution by **+1.2** points over contrastive-only training. Furthermore, at resolution 1036 the student reaches a **2.64×** speedup over ViT-Distill.

## 6 LIMITATION AND FUTURE WORK

While C-GSPN delivers clear gains in high-resolution scalability by replacing attention with Latent 2D Linear Spatial Propagation, the block's feed-forward MLP remains unmodified. Profiling at batch size 32 indicates that, at resolutions $\geq 512$, the MLP accounts for over 52% of the total C-GSPN block latency. Future work will focus on targeted MLP optimization, including compression, kernel fusion, and low-rank variants to unlock additional end-to-end speedups.

## 7 CONCLUSION

We introduced C-GSPN, a compact spatial propagation block that replaces attention in ViTs. Propagating in a compressed latent space with fused CUDA normalization and distilled via dual PP/PB taps, C-GSPN delivers large high-resolution speedups without tiling while maintaining competitive zero-shot accuracy and stronger dense prediction, offering a practical path toward subquadratic foundation vision encoders.

**Ethics Statement** This work adheres to the ICLR Code of Ethics. In this study, no human subjects or animal experimentation was involved. All datasets used, including DataComp, Ade20K, Pascal, COCO, were sourced in compliance with relevant usage guidelines, ensuring no violation of privacy. We have taken care to avoid any biases or discriminatory outcomes in our research process. No personally identifiable information was used, and no experiments were conducted that could raise privacy or security concerns. We are committed to maintaining transparency and integrity throughout the research process.

**Reproducibility Statement** We have made every effort to ensure that the results presented in this paper are reproducible. The experimental setup, including hyperparameters and hardware details, is described in detail in the paper. Additionally, all datasets used in the paper are publicly available, ensuring consistent and reproducible evaluation results. We will release all code upon publication to enable replication and verification.

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

TABLE OF CONTENTS

## A  ADDITIONAL RELATED WORK

### A.1  SUBQUADRATIC ATTENTION AND ALTERNATIVES

Subquadratic alternatives to full softmax attention aim to reduce the $\mathcal{O}(N^2)$ dependence on token count while preserving global interactions. *Sparsity- and window-based* designs such as Longformer and BigBird in NLP and Swin for vision constrain attention to local windows with a few global tokens, yielding near-linear scaling but making long-range mixing sensitive to the chosen sparsity pattern and hyperparameters (Beltagy et al., 2020; Zaheer et al., 2020; Liu et al., 2021). *Kernelized/low-rank* approaches linearize attention—e.g., Linear Transformers, Performer, Nyströmformer, Linformer—trading exactness for approximation; their accuracy often depends on the feature map, rank, or landmark scheme and may require careful tuning (Katharopoulos et al., 2020; Choromanski et al., 2020; Xiong et al., 2021; Wang et al., 2020). IO-aware exact attention like FlashAttention reduces constant factors via optimized memory access, yet latency still scales quadratically with tokens at high resolution (Dao et al., 2022). Beyond attention, *state-space models* (S4; Mamba) offer linear-time sequence operators, but adapting 1D formulations to high-resolution vision typically requires extra 2D inductive bias or hierarchical designs (Gu et al., 2021; Gu & Dao, 2023). In contrast, *spatial propagation networks* operate natively on 2D grids and remove positional embeddings; recent GSPN extends this idea to four-direction propagation with linear complexity (Liu et al., 2017; Wang et al., 2025). Our work scales GSPN to foundation-model pretraining through a compact, CUDA-optimized instantiation distilled from ViT teachers, achieving substantially lower latency at 1K–2K while maintaining competitive transfer.

### A.2  FOUNDATION MODEL DISTILLATION

Knowledge distillation (KD) has long been employed to compress large models into more efficient students. Early works explored attention transfer in CNNs (Zagoruyko & Komodakis, 2017), while DeiT (Touvron et al., 2021) demonstrated that ViT-to-ViT distillation can achieve strong performance at scale, highlighting the potential of KD for transformers. Subsequent studies refined these ideas: ViTKD guidelines (Yang et al., 2022) emphasized the importance of intermediate supervision and careful layer alignment for stable training. More recently, KD research has extended beyond isomorphic student–teacher pairs to span across operator families. For instance, quadratic-to-subquadratic transfer has been explored to compress attention-heavy architectures into efficient

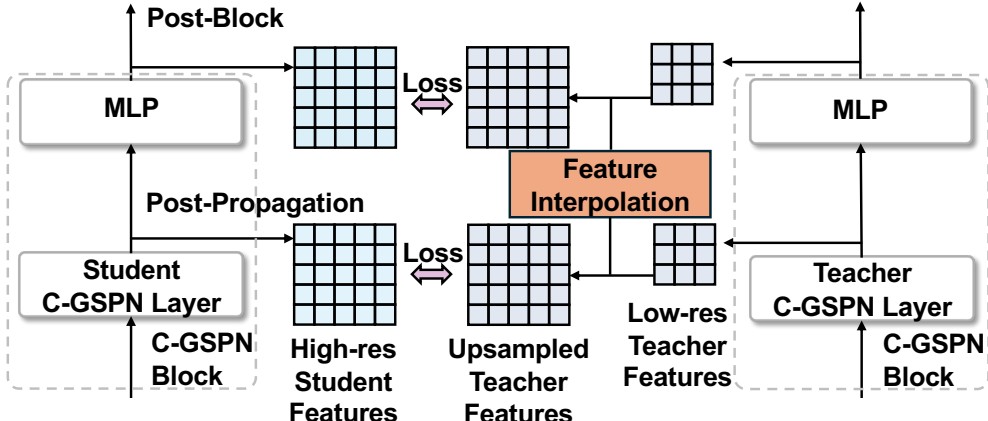

Figure 7: High-resolution encoder distillation: a frozen low-resolution teacher supervises a higher-resolution student via upsampled features at two taps (post-propagation and post-block), with feature interpolation bridging resolutions and applied progressively in a resolution curriculum. See Sec. 4.3 in the main paper for details.

approximations (Bick et al., 2024), and hybrid schemes distilling Mamba into Transformer backbones have emerged (Li et al., 2025). These works suggest that KD can serve as a bridge across heterogeneous operator classes, though most prior efforts remain at modest scales or focus on small task-specific settings. In contrast, our work investigates *foundation-scale distillation across operator families*, targeting the challenging scenario of attention-to-propagation transfer. We demonstrate that staged supervision (layer-wise and block-level) combined with CUDA-optimized latent propagation enables both competitive accuracy and substantial efficiency gains, scaling effectively to high-resolution inputs (1K–2K) while preserving transfer performance.

# B  GPU HARDWARE AND KERNEL EXECUTION FOR 2D LINEAR PROPAGATION

Modern GPUs, such as NVIDIA's A100, enable high parallelism through a hierarchical execution model involving grids, thread blocks, and warps. A kernel—a compiled function for GPU execution—is launched as a grid of thread blocks, where each block contains up to 1024 threads organized into 32-thread warps, the basic scheduling unit on streaming multiprocessors (SMs; 108 on A100). Warps execute in a single-instruction, multiple-thread (SIMT) manner, maximizing throughput when occupancy—the proportion of active warps per SM—is high, balanced against constraints like register usage (up to 65,536 per SM) and shared memory (up to 164 KB per SM).

In sequence modeling architectures like 2D linear propagation (Wang et al., 2025; Liu et al., 2017), input tensors of shape $B \times C \times H \times W$ (batch size $B$, channels $C$, height $H$, width $W$) are processed via a line-scan propagation scheme. This involves sequential row or column updates with parallel computations within each step. The CUDA implementation maps spatial dimensions ($H \times W$) to threads, while $B$ and $C$ define independent slices for concurrent processing. In the kernel, a 1D block configuration might allocate blockDim.x to a fixed number of threads (e.g., 512), with the grid size scaled by $B \times C \times H$ (or $B \times C \times W$) to distribute the workload across SMs. Each thread handles a pixel along the parallel spatial axis, launching a separate kernel per propagation step (e.g., per row or column), which results in thousands of micro-launches. This design, however, faces scalability challenges with large $B \times C$. GPUs have finite concurrency limits, constrained by the number of SMs and per-SM block capacity (32 blocks). When $B \times C$ exceeds these limits, excess slices are processed sequentially, causing runtime spikes despite the theoretical parallelism.

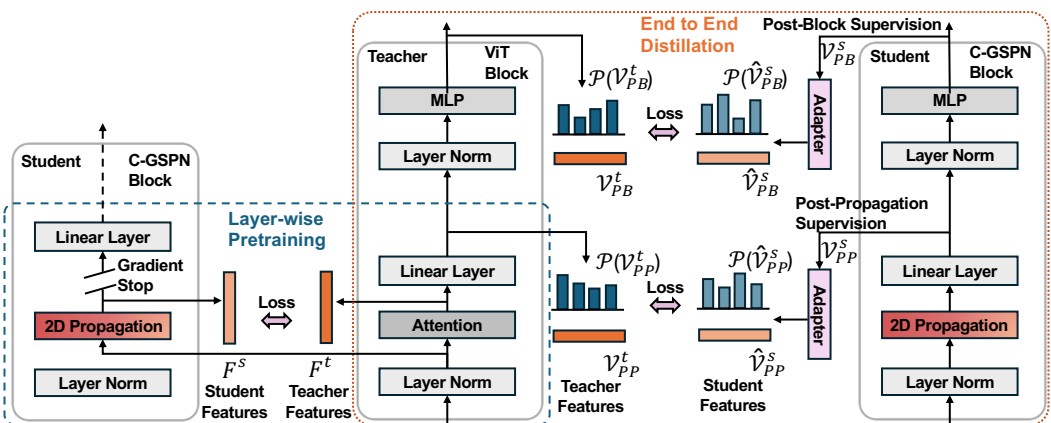

Figure 8: **Two-stage distillation for scaling C-GSPN**. Stage 1: Sublayer-wise pretraining aligns each C-GSPN propagation sublayer to the teacher's attention sublayer. Stage 2: End-to-end distillation applies dual taps—post-propagation (PP) and post-block (PB)—with lightweight feature adaptors to reduce feature-space mismatch.

## C  IMPLEMENTATION DETAILS

### C.1  PRETRAINING

Before initiating end-to-end distillation, we conduct a lightweight pretraining stage designed to stabilize optimization and provide a strong initialization. Specifically, we train on 5M image–text pairs sampled from the DataComp benchmark (Gadre et al., 2023), which balances diversity and scale. We adopt the AdamW optimizer (Loshchilov & Hutter, 2017) with a learning rate of $4 \times 10^{-5}$, a global batch size of 1024, and 300 warmup steps. The schedule follows linear decay, gradually annealing the learning rate to zero. This setup encourages early convergence without overfitting, and the pretrained weights serve as a robust initialization for subsequent supervised distillation. Our empirical analysis shows that omitting this step leads to unstable training in the early epochs and consistently lower downstream performance.

### C.2  END-TO-END DISTILLATION TRAINING

For full-scale training, we distill C-GSPN on 600M curated image–text pairs from DataComp. The student model is optimized to align with its teacher (OpenCLIP SO/14) through staged supervision, as outlined in Section 4.2. We adopt a sparse distillation strategy, where we only distill every ninth block of the teacher model. We again use AdamW with a higher learning rate of $4 \times 10^{-4}$, a global batch size of 8192, and a cosine decay learning-rate schedule with $10\,000$ warmup steps. This configuration provides both the stability required for large-batch training and the flexibility to adapt across the different supervision stages.

### C.3  LOSS COMPOSITION AND BALANCING

The total distillation loss combines the two supervision taps per block—*post-propagation (PP)* and *post-block (PB)*:

$$\mathcal{L} = \alpha\,\mathcal{L}_{\text{PP}} + \beta\,\mathcal{L}_{\text{PB}}, \tag{15}$$

with

$$\mathcal{L}_{\text{PP}} = \text{MSE}\big(V_{\text{PP}}^s, V_{\text{PP}}^t\big) + \lambda_1\,\text{KL}\big(P(V_{\text{PP}}^s) \,\|\, P(V_{\text{PP}}^t)\big), \tag{16}$$

$$\mathcal{L}_{\text{PB}} = \text{MSE}\big(V_{\text{PB}}^s, V_{\text{PB}}^t\big) + \lambda_2\,\text{KL}\big(P(V_{\text{PB}}^s) \,\|\, P(V_{\text{PB}}^t)\big).$$

Here, $V_{\text{PP}}^{s/t}$ and $V_{\text{PB}}^{s/t}$ denote student/teacher features at the PP and PB taps, and $P(\cdot)$ is the token-wise softmax distribution. We set $\alpha = \beta = 0.5$ to balance PP and PB supervision, ensuring that the propagation sublayer is directly constrained without being overshadowed by block-level matching.

| Dataset | 378-teacher | 378-multires | 448-multires | 518-multires |
|---------|-------------|--------------|--------------|--------------|
| ADE20K  | 46.0        | 45.8         | 45.8         | 45.9         |

Table 3: Multi-resolution distillation on ADE20K (mIoU). A single student trained to support multiple input resolutions matches the single-resolution baseline.

The divergence weights $\lambda_1 = \lambda_2 = 7/3$ provide a balance between feature-level alignment (MSE) and distributional matching (KL). To reduce feature-space mismatch, a lightweight 2-layer MLP adaptor is inserted before each tap (Sec. 4.2).

### C.4 STABILITY PRACTICES

Layer-wise pretraining (Stage 1) provides consistent signals to each sublayer before end-to-end optimization (Stage 2). In ablations, removing either the adaptors or Stage 1 degrades stability and final accuracy.

## D MORE EXPERIMENTAL RESULTS

We evaluate multi-resolution distillation by training a single C-GSPN model that operates across multiple input resolutions without special positional embeddings. A low-resolution teacher supervises a multi-resolution student during distillation. As shown in Table 3, the student maintains comparable performance across 378, 448, and 518 resolutions, indicating that our approach transfers effectively across scales.

## E THE USE OF LARGE LANGUAGE MODELS (LLMS)

We used large language models (OpenAI GPT) only for wording suggestions. All outputs were reviewed and edited by the authors; no analyses, results, or code central to our contributions were generated by LLMs, and no sensitive data were provided.

