# OpenReview forum: "Compact GSPN: Scaling Spatial Propagation to Vision Foundation Models"
_ICLR.cc/2026/Conference — ICLR 2026 Conference Withdrawn Submission_

### Official Review · Reviewer_MoPC · 2025-10-30

**Soundness:** 2
**Presentation:** 3
**Contribution:** 2
**Rating:** 4
**Confidence:** 3

**Summary:**

In this work, the authors proposed compact GSPN, a new method based on GSPN (Generalized Spatial Propagation Networks) that further reduce the complexity and improve the efficiency. Two solutions are provided for the efficiency: lightweight projections and fused CUDA kernels. To train at foundation model scale, the authors consider a two-stage cross-operator distillation recipe: first layer-wise distillation supervision and then end-to-end fine-tune. Extensive experiments show strong performance.

**Strengths:**

1. Extensive experiments show good performance of proposed method.
2. Good presentation make the paper easy to follow.
3. Detailed analysis and designs.
4. While reader may not familar with GSPN, a background section is a good choice to introduce some background.

**Weaknesses:**

1. I would suggest not emphasize Fig.4  considering minimal contribution here. Meanwhile, reducing the channel dimension must lead to performance drop, empirically. How the authors handle this issue or how to balance the efficiency and performance.

2. While the authors claimed and reported efficiency mainly for 1k resolution, but the main results reported in Table 1 are achieved with 384x384 resolution. I assume in this resolution, the proposed method cannot imporve the performance a lot? Better to report the on-device  (GPU or CPU or mobile) latency in Table 1 for all methods. This can help better understand the balance of performance and efficiency.

3. In fig. 3, it is clear remove components from the model can speed up, but what about the performance drop? Better to also provide evaluation results in Fig 3 after removing each component.

4. Why we need layer-wise distillation as shown in L238?

5. I did not get the motivation doing  sublayer-wise Pretraining. The proposed method and SigLIP2 are fundementally different architecture, what is the motivation doing sublayer-wise Pretraining (i.e., aligns intermediate features block by block)? Or if any empirical results indicate that this can provide a better performance under same total computational bugets.

6. Why consider Eq. 9 as the loss for layer-wise guidance? Enforcing block output same for different architectures makes no sense to me.
And what is the motivation for " every C-GSPN sublayer directly learns to mimic the representational pattern of its paired attention sublayer"?

7. Also, the  End-to-end Distillation seems not reasonable to me. Aligning features "after the propagation/attention sublayer" and  "after the entire block" is not clear. Even do layer-wise, why not just the final output for each layer (say PB in the paper).

**Questions:**

1. For the four directions scan, top-down, down-top, left-right,  and right-left, can we reduce to two? Meanwhile, scanning from different directions and combine these results might not be a good and elegent solution for vision, like many Mamba-applications for vision.

2. I'm not very familiar with CUDA kernel, hence I will have no detailed suggestions and questions here. I will follow other reviewers' comments.

3. For the section Fused CUDA Normalization, does that mean we can also do CUDAv optimization for other models like VIT, and other models can also be improved? and what this the relationship between Fused CUDA Normalization and the main idea in this paper?

---

### Official Review · Reviewer_Cw9U · 2025-10-31

**Soundness:** 3
**Presentation:** 2
**Contribution:** 2
**Rating:** 4
**Confidence:** 3

**Summary:**

This paper follows a previous work "GSPN" and proposes a compact vision of it for scaling-up scenario, especially for vision backbone. By using a series of designed approaches, this Compact GSPN outperforms other comparsion baselines.

Based on details below, here is a jusitification of my initial decision.
Since I am not very familiar with the vision backbone efficiency design, my score is more like a borderline, but the system does not let me do so. I will check the other reviewers' comments to adjust my final score. The author can simply treat my current score as "5".

**Strengths:**

1. The motivation itself is solid, the efficiency issue exists in the vision backbone, which limits the potential useage for very high resolution visual signal process.
2. Even if I give "2" for presentation, but I think the writing itself lets the reader following the draft. The reasons for 2 is the method itself is not easy to present, which contains several steps.
3. Different designs parts are relied on different motivation and experimental observations, which are solid.
4. Performances are good compared with baselines.

**Weaknesses:**

1. Some parts of presentation may be improved. ex. I suggest the author supplement the related work section, at least cover all necessary intros in the main draft to make it complete. You could definitely supplement more in the appendix for each part.
2. Performance gain is not that significant, I am not sure if such level improvement is good enough for backbone efficiency field.
3. Since this work is following GSPN directly, as shown in the figure, I will not judge the novelty by "following a previous work", but the newly involved parts look straightforward to me. Overall this series of techniques is more like a very detailed empirical analysis for engineering. I concern about the research paper novelty while I still value the technical contribution.
4. As 1 above, some figures need improvement. Like fig.5, these lines are too close to each other, the confusion part is, you already indicated a legend, but why also using arrow to point out the name of each method?

**Questions:**

Please check above section.

---

### Official Review · Reviewer_ubhX · 2025-10-31

**Soundness:** 3
**Presentation:** 2
**Contribution:** 2
**Rating:** 4
**Confidence:** 4

**Summary:**

The paper proposes Compact GSPN (C-GSPN), replacing ViT attention with a latent-space 2D propagation block plus a fused CUDA normalization kernel, and a two-stage cross-operator distillation (layer-wise, then end-to-end with dual taps at post-propagation and post-block). The goal is to keep transfer performance while cutting latency at high resolutions. Reported results show large kernel/block speedups and competitive downstream accuracy; segmentation even improves over the ViT teacher in some settings.

**Strengths:**

- Practical CUDA analysis:
The paper’s analysis of model parallelism on real CUDA hardware is valuable. The authors go beyond abstract complexity and examine the actual GPU concurrency limits that slow GSPN propagation. This kind of system-level insight is rarely presented in CV papers and provides useful reference for readers who may not be familiar with CUDA kernel bottlenecks or SM utilization issues.

- Engineering-driven efficiency:
The proposed improvements, including latent-space propagation and kernel fusion, show significant practical acceleration. Although part of the gain comes from engineering effort rather than algorithmic novelty, these optimizations are concrete, reproducible ideas that can benefit future efficient vision architectures.

**Weaknesses:**

- Justification of low-rank latent propagation (L161):
The paper assumes that the latent-space representation can be compressed along channels with little loss, but it does not quantify this assumption. Showing the channel-wise correlation or energy spectrum of features would provide empirical justification for using low-rank approximation.

- Missing architecture overview (L188):
The description of the C-GSPN architecture is scattered across sections. It will be good to have a architecture figure to show which parts are the non-propagation modules.

- Fused CUDA normalization details (L204):
Since much of the speed improvement comes from engineering, more implementation detail is needed. A pseudo-code snippet or release of the kernel (even in supplementary) would help others reproduce the claimed gains.

- Layer-wise pretraining (L254):
The “frozen teacher / per-layer student” stage resembles early 2010s layer-wise training. The authors should clarify whether this is only an initialization step and show ablations proving its necessity, as modern distillation typically relies on full end-to-end optimization.

- Feature adaptor design and prior work (L294):
Feature adaptors are widely used in existing distillation methods, e.g., FitNet, DeiT, and other distillation work for detection and segmentation. The paper should explicitly position its adaptor design relative to these works and explain what is new about its two-tap formulation.

- Hybrid architecture clarity (L317):
The paper mentions a “hybrid” with some attention layers preserved but does not specify where or how many. A structural diagram or ablation of attention placement would make this design easier to follow.

- Scaling to different model sizes:
The original GSPN reports results for Tiny/Small/Base variants. It would be informative to see how the proposed modifications affect models at different scales, as efficiency techniques sometimes behave inconsistently with capacity.

- Applicability to heterogeneous backbones:
The proposed distillation approach could theoretically apply to other architecture pairs, e.g., CNN→ViT.  Demonstrating such generality, even on a small benchmark, would strengthen the paper’s impact, for example, I am curious to see an experiments on ViT -> ConvNext distillation using the proposed method.

- Accuracy trade-off (Table 1):
The proposed C-GSPN slightly reduces classification accuracy compared with the ViT-Distill teacher while improving segmentation results for a bit. The paper should discuss this trade-off explicitly instead of framing the method as universally superior.

- Insufficient ablation depth:
There is limited analysis on how each proposed module (latent compression, fused norm, hybrid attention, and distillation) contributes individually to accuracy and speed. More controlled ablations and multi-seed runs would make the conclusions more convincing.

- Limited novelty:
  - the C-GSPN block is all about "hey, doing the propagation with original dimension is expensive, let's reduce the dimension". Also the idea of predicting propagation parameters from the features maps is very similar to Mamba
  - Distillation is very hackary, training a network layer by layer is certainly not what people need in the year of 2025
  - The curriculum learning idea in resolution transfer is also from previous works, e.g.,  Li et al., 2024

**Questions:**

No additional question.

---

### Note · Authors · 2025-11-14

I have read and agree with the venue's withdrawal policy on behalf of myself and my co-authors.